# Application of Polyaluminium Chloride Coagulant in Urban River Water Treatment Influenced the Microbial Community in River Sediment

**Siyu Liu** [1,†], **Xuchao Zhuang** [2,†] **and Chuandong Wang** [1,*]

[1] State Key Laboratory of Microbial Technology, Microbial Technology Institute, Shandong University, Qingdao 266237, China; liusy523@mail.sdu.edu.cn

[2] Hangzhou Keyun Environmental Technology Co., Ltd., Hangzhou 310000, China; zxc8844@126.com

[*] Correspondence: wangchuandong@sdu.edu.cn; Tel.: +86-532-5863-1538

[†] These authors contributed equally to this work.

**Abstract:** Polyaluminium chloride (PAC) has been widely used as a chemical coagulant in water treatment. However, little is known about the impact of PAC performance on the microbial community in sediments. In this study, the archaeal, bacterial, and fungal communities in urban river sediments with and without PAC treatment were investigated. Prokaryotic diversity decreased at the PAC addition site (A2) and increased along with the river flow (from A3 to A4), while eukaryotic diversity was the opposite. The abundance of core microbiota showed a similar trend. For example, the dominant Proteobacteria presented the highest relative abundance in A1 (26.8%) and the lowest in A2 (15.3%), followed by A3 (17.5%) and A4 (23.0%). In contrast, Rozellomycota was more dominant in A2 (56.6%) and A3 (58.1%) than in A1 (6.2%) and A4 (16.3%). Salinity, total dissolved solids, and metal contents were identified as the key physicochemical factors affecting the assembly of core microorganisms. The predicted functions of archaea and fungi were mainly divided into methane cycling and saprotrophic nutrition, respectively, while bacterial function was more diversified. The above findings are helpful to enhance our understanding of microorganism response to PAC and have significance for water treatment within the framework of microecology.

**Keywords:** polyaluminium chloride coagulant; urban river sediment; microbial community; function prediction

## 1. Introduction

As a part of surface water, urban rivers and streams are the most critical human water resources; they are essential for a wide range of fields, such as irrigation, animal husbandry, energy, manufacturing, and household use [1]. Urban rivers and riverine environments are defining features of many cities with increasing population growth, affecting the happiness of urban residents [2]. However, uncontrolled urbanization and industrial expansion have seriously threatened the freshwater ecosystem over the years [3,4]. The pollutants originating from untreated industrial, agricultural, and domestic wastewater are linked to water quality deterioration and sometimes cause irreparable damage to the global environment [5–7]. Declining water quality can cause eutrophication, sedimentation, harmful algal blooms, hypoxia, and dead zones, all of which negatively impact social and economic development and the maintenance of a healthy ecosystem [7]. Although recent improvements have been made in providing clean drinking water, surface water is still considered one of the most degraded natural resources in China [8]. With the rapid development of industrialization and urbanization, the discharge of industrial and domestic sewage has led to an increase in the load of nutrients, metals, and pesticides in urban surface runoff. These changes affect river ecosystems and pose threats and challenges to the sustainable use of rivers [9].

The Beitang River is located in Hangzhou City, Zhejiang Province, China. It is 36 km long, 30–35 m wide, and 2–2.5 m deep from west to east. Since its establishment in 1977, this river has been used for agricultural water conservancy irrigation and river transportation. Part of the Beitang River's water flows into the Xixinghou River through underground pipes, and the latter becomes a scenic river in the urban area. In recent years, the Beitang River has been seriously polluted, mainly caused by sewers that are not intercepted, domestic garbage, and wastewater from enterprises [10], resulting in a severe decline in the water quality of the Xixinghou River. As the Xixinghou River contains many suspended particles, coagulation/flocculation (CF) treatment is performed by adding polyaluminium chloride (PAC) coagulant to improve water quality. The CF treatment is a well-established physicochemical process with the advantages of low cost and easy operation, widely used in the purification of water and wastewater [11]. So far, trivalent aluminum-based compounds or polymers remain one of the most commonly used coagulants for removing extracellular organic matter [12], dye [13], particles [14], and heavy metal cations [15] since most of the suspended colloids in water are usually negatively charged. Compared with traditional hydrolyzed aluminum salt, PAC is frequently selected as coagulants because of its effective charge neutralization [16] and better performance at low temperatures [17].

It is well-understood that microorganisms inhabiting the aquatic environment drive crucial ecosystem processes and contribute to global biogeochemical cycles [18], the biodegradation and biotransformation of pollutants [19], and the restoration and maintenance of aquatic ecosystems [20]. The microbial communities exhibit taxonomically distinct and functional diversity at different spatial scales in urban river systems due to natural and anthropogenic activities [21,22]. PAC has been widely used in water treatment. However, to date, whether the application of PAC affects aquatic microecology is still little known. Therefore, it is quite necessary to investigate the shift of microbial communities in response to PAC application. In this study, a culture-independent approach was carried out to compare archaeal, bacterial, and fungal community structures in urban river sediments treated with or without PAC coagulant by sequencing 16S rRNA and internal transcribed spacer (ITS) sequences of distinct regions. It is hoped that this study would provide an explorative and descriptive reference for the development of PAC in urban river management from the perspective of microbial community shift.

## 2. Materials and Methods

### 2.1. Sample Collection and Genomic DNA Extraction

One sampling site (A1) was selected for the Beitang River, and three sampling sites (A2, A3, A4) were selected along the flow direction of the Xixinghou River (Figure 1a). PAC was added into the river at site A2 for CF treatment. Four sediment samples were stored at −80 °C with sterilized equipment. Total genomic DNA was extracted from 0.5 g samples using the FastDNA SPIN Kit for Soil (MP Biomedicals LLC., Santa Ana, CA, USA) according to the manufacturer's instructions [23]. The purity and concentration of the extracted genomic DNA were monitored on 1% agarose gels and evaluated using an ND-1000 NanoDrop spectrophotometer (Thermo Fisher Scientific Inc., Wilmington, NC, USA). According to the measured concentration, the genomic DNA was diluted to 1 ng/μL using sterile water.

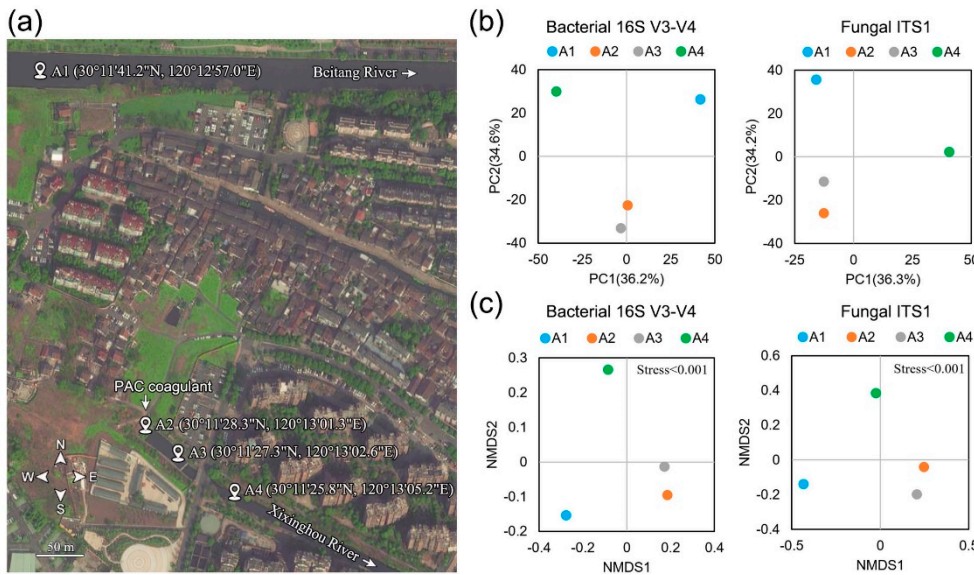

**Figure 1.** Biogeography characteristics of the bacterial and fungal communities in the sampled river sediments. (**a**) Map showing the location and distribution of sampling sites (A1–A4) across the urban region of the Beitang River and the Xixinghou River. The white arrow indicates the flow direction. (**b**) PCA and (**c**) NMDS plots of bacterial and fungal communities in the sediment samples based on the 97% similarity of OTUs.

## 2.2. Physicochemical Analysis of Sampling Sites

Turbidity was measured using a portable turbidity meter (2100Q, Hach Company, Loveland, CO, USA) and expressed in nephelometric turbidity units (NTUs). Total dissolved solids (TDS) and salinity were determined by a microprocessor conductivity meter (DDB-350F, INESA Scientific Instrument Co., Ltd., Shanghai, China), as described previously [24]. The pH was monitored by using a PB-10 pH meter (Sartorius, Göttingen, Germany). The C/N ratio was calculated based on the carbon and nitrogen content in the sample, determined by a micro elemental analyzer (UNICUBE, Elementar, Langenselbold, Germany). The metal concentrations were detected by inductively coupled plasma mass spectrometry (NexION-1000G, PerkinElmer, Waltham, MA, USA). All measurements were performed in triplicate.

## 2.3. Gene Amplicons and Illumina Miseq Sequencing

16S rRNA and ITS sequences of distinct regions (16S V3-V4, ITS1) were amplified using specific primers with a barcode. The forward primer 338F (5′-ACTCCTACGGGAGGCAG CAG-3′) and the reverse primer 806R (5′-GGACTACHVGGGTWTCTAAT-3′) were used to amplify the V3-V4 region of 16S rRNA genes [25]. The ITS1 region of fungal nuclear DNA was amplified with primers ITS1F (5′-CTTGGTCATTTAGAGGAAGTAA-3′) and ITS2 (5′-GCTGCGTTCTTCATCGATGC-3′) [26]. For the amplification, the 20-μL reaction mix contained 2 × Phanta® Max Buffer, 10 mM dNTP Mix, 1 μM of each specific primer (as described), 0.5 U of Phanta® Max Super-Fidelity DNA polymerase (Vazyme, Nanjing, China), and 10 ng template DNA. Thermal cycling consisted of initial denaturation at 98 °C for 2 min, followed by 30 cycles of denaturation at 98 °C for 10 s, annealing at 50 °C for 30 s, elongation at 72 °C for 30 s, and a final extension at 72 °C for 5 min. Sequencing libraries were generated by a TruSeq® DNA PCR-Free Sample Preparation Kit (Illumina, Inc., San Diego, CA, USA) following the manufacturer's recommendations and then sequenced on an Illumina NovaSeq platform (Novogene Bioinformatics Technology Co., Ltd., Beijing, China).

### 2.4. Sequencing Data Processing and Bioinformatics Analysis

Paired-end reads were assigned to samples based on their unique barcode and truncated by cutting off the barcode and primer sequence. Forward and reverse sequences were merged by overlapping paired-end reads using Fast Length Adjustment of SHort reads (FLASH, V1.2.7, http://ccb.jhu.edu/software/FLASH/, accessed on 4 August 2013) [27]. In terms of quality control, quality filtering on raw tags was performed under Quantitative Insights Into Microbial Ecology (QIIME) (V1.9.1, http://qiime.org/scripts/split_libraries_fastq.html, accessed on 1 January 2018) to obtain the high-quality clean tags [28]. Tags were compared with the Silva database (Release138, https://www.arb-silva.de/, accessed on 27 August 2020) by the UCHIME algorithm (http://www.drive5.com/usearch/manual/uchime_algo.html, accessed on 30 March 2020) to detect and remove off chimera sequences [29]. Sequences of over 97% similarity were considered the same operational taxonomic units (OTUs), and a representative sequence for each OTU was screened for further annotation. Alpha diversity indices, including Good's coverage, Shannon, Simpson, Chao1, and abundance-based coverage estimator (ACE), were calculated by QIIME to show the species complexity of one sample, while beta diversity analysis was conducted for evaluating sample differences in species complexity. Principal component analysis (PCA) and non-metric multidimensional scaling (NMDS) analysis based on weighted UniFrac distances were performed to visualize broad trends of similarities and differences of the related samples. The relationship between species diversity and environmental variables was analyzed using redundancy analysis (RDA). Based on the sequencing data, the Functional Annotation of Prokaryotic Taxa (FAPROTAX, http://www.loucalab.com/archive/FAPROTAX, accessed on 27 November 2020) [30] and FUNGuild (https://github.com/UMNFuN/FUNGuild, accessed on 25 July 2017) [31] databases were used for predictive functional analysis of bacterial and fungal communities.

### 2.5. Sequence Accession Number

The raw data reported in this article are available in the NCBI Sequence Read Archive (SRA, https://trace.ncbi.nlm.nih.gov/Traces/sra/sra.cgi, accessed on 21 March 2021) under BioProject accession number PRJNA716078.

## 3. Results

### 3.1. Physicochemical Properties of Sampling Sites

The physicochemical properties of surface water and river sediments are partially listed in Table 1. The sampling sites were 0.80 to 2.40 m deep and showed weak alkalinity, with a pH ranging from 7.23 to 7.51. Sampling site A1 is located in the Beitang River (Figure 1a), which is upstream of the Xixinghou River and appears turbid ($315 \pm 11$ NTUs). CF treatment with PAC coagulant was carried out at the site of A2, which resulted in a decrease in turbidity ($280 \pm 9.80$ NTUs) and an increase in TDS ($488 \pm 7.72$ mg/L) and salinity ($0.505 \pm 0.0210$ g/kg). Compared with A1 and A2, the turbidity values of A3 ($5.91 \pm 1.48$ NTUs) and A4 ($3.67 \pm 0.778$ NTUs), located downstream of the Xixinghou River, decreased significantly ($p < 0.01$). The TDS and salinity of A3 reduced, while those values increased in A4. The value of the C/N ratio in sediments was the highest in A1 ($22.1 \pm 1.91$) and the lowest in A3 ($9.20 \pm 1.41$). The major metal elements observed in the sediments from the four sampling sites were magnesium (Mg), aluminum (Al), calcium (Ca), and manganese (Mn) (Table 1). It was found that the contents of Mg, Al, Ca, and Mn in A2 was $23.7 \pm 1.01$, $0.158 \pm 0.00213$, $1.16 \times 10^3 \pm 17.6$, and $5.27 \pm 0.0162$ mg/L, respectively, which were all higher than those in upstream A1 (Mg $20.2 \pm 1.33$, Al $0.0729 \pm 0.00125$, Ca $647 \pm 10.1$, and Mn $0.0185 \pm 0.00244$ mg/L). The contents of Mg, Ca, and Mn all decreased in A3 and increased in A4, while the content of Al increased from $0.0729 \pm 0.00125$ to $0.599 \pm 0.00275$ mg/L along the flow direction of the Xixinghou River.

**Table 1.** Physicochemical properties and partial metal contents of the sampling sites and sediment samples.

| Sample ID | Depth (m) | Turbidity (NTUs) [a] | TDS [b] (mg/L) | Salinity (g/kg) | pH | C/N [c] Ratio | Total Mg (mg/L) | Total Al (mg/L) | Total Ca (mg/L) | Total Mn (mg/L) |
|---|---|---|---|---|---|---|---|---|---|---|
| A1 | 2.00–2.20 | 315 ± 10.6 | 426 ± 12.4 | 0.447 ± 0.0350 | 7.51 ± 0.0208 | 22.1 ± 1.91 | 20.2 ± 1.33 | 0.0729 ± 0.00125 | 647 ± 10.1 | 0.0185 ± 0.00244 |
| A2 | 0.80–1.00 | 280 ± 9.80 | 488 ± 7.72 | 0.505 ± 0.0210 | 7.24 ± 0.0186 | 19.4 ± 1.45 | 23.7 ± 1.01 | 0.158 ± 0.00213 | $1.16 \times 10^3$ ± 17.6 | 5.27 ± 0.0162 |
| A3 | 1.50–1.80 | 5.91 ± 1.48 | 349 ± 2.10 | 0.378 ± 0.0140 | 7.23 ± 0.0171 | 9.20 ± 1.41 | 13.3 ± 0.316 | 0.177 ± 0.00180 | 870 ± 15.1 | 0.0166 ± 0.00120 |
| A4 | 2.10–2.40 | 3.67 ± 0.778 | 457 ± 5.46 | 0.493 ± 0.0170 | 7.47 ± 0.0386 | 12.9 ± 3.30 | 21.6 ± 0.622 | 0.599 ± 0.00275 | $1.24 \times 10^3$ ± 20.6 | 1.08 ± 0.00701 |

[a] NTUs, nephelometric turbidity units. [b] TDS, total dissolved solids. [c] C/N ratio, carbon to nitrogen ratio.

### 3.2. Microbial Diversity and Abundance

After the quality-filtered and chimera-checked process of raw sequencing reads, a total of 150,886 high-quality bacterial sequences and 249,643 high-quality fungal sequences from the four sediment samples were identified with an average length of 419.5 and 237.25 nucleotides (Table S1). On average, 1467 bacterial OTUs and 766 fungal OTUs were obtained with a 97% identity threshold, respectively (Table S1). To estimate the species richness in samples with different sequencing data and indicate the rationality of the amount of sequencing data, the rarefaction curves at 97% similarity grouping of the 16S V3-V4 and ITS1 sequences were plotted in Figure S1. The rarefaction curves of bacteria and fungi tended to be flat with the increase of sequencing data, indicating that the sequencing depth was adequate to capture most of the bacterial and fungal diversity in each sample. Good's coverage estimates showed that 99.1% of the bacterial species and 99.8% of the fungal species were correctly covered by current microbial profiles (Table S2). Two indices of alpha diversity, including Shannon and Simpson, were calculated to represent species diversity, while Chao1 and ACE were calculated for species richness (Table S2). Shannon and Simpson estimators showed that bacterial community diversity in A1 (8.12, 0.02) and A4 (8.19, 0.02) were higher than those in A2 (7.51, 0.03) and A3 (7.33, 0.05), and fungal community diversity was the lowest in A1 (3.23, 0.39) and highest in A3 (5.04, 0.08). The Chao 1 and ACE estimators showed that bacterial community abundance was highest in A1 (1631.01, 1661.42) and weakest in A2 (1564.86, 1553.88). In contrast, fungal community abundance was highest in A4 (920.20, 941.39) and lowest in A3 (737.82, 757.52). In order to clarify the beta diversity separation of the microbial community structure based on the sampling locations, statistical methods PCA (Figure 1b) and NMDS (Figure 1c) were conducted to analyze the variations in bacterial and fungal communities before (A1), during (A2), and after the application of PAC (A3 and A4). The PCA analysis results showed that the two axes explained 70.8% (PC1 36.2% and PC2 34.6%) of the bacterial variation and 70.5% (PC1 36.3% and PC2 34.2%) of the fungal variation in species, respectively (Figure 1b). The samples A2 and A3 showed a homologous cluster and were differentiated from the A1 and A4 samples. The NMDS analysis displayed similar results (Figure 1c). Generally, these results suggested that the samples A1 and A4 had a distinctive microflora cluster, whereas the microflora from the sites of A2 and A3 were analogous.

### 3.3. Variations of Microbial Composition

The microbial composition varied among the different sampling sites. A total of 6 archaeal phyla was identified in all the sediment samples, including Crenarchaeota, Euryarchaeota, Halobacterota, Micrarchaeota, Nanoarchaeota, and Thermoplasmatota (Figure 2a). The most abundant phylotype belonged to Halobacterota, ranging from 60.8% to 88.7%, followed by Nanoarchaeota (1.1% to 20.6%) and Crenarchaeota (1.0% to 8.4%) (Table S3). The bacterial community was dominated by the phyla Proteobacteria, Firmicutes, Cyanobacteria, Bacteroidota, and Nitrospirota (Figure 2b). Proteobacteria was the most abundant taxon in A1 (26.8%) and A4 (23.0%), whereas Firmicutes became predominant in A2 (23.4%) and A3 (23.2%) (Table S4). The phyla Nitrospirota was present at the highest relative abundance in A2 (15.8%) compared with the other samples (1.1% to 4.8%). The distribution of fungal phyla for the OTUs showed that the sequence reads could mainly be classified in Rozellomycota, Chytridiomycota, Basidiomycota, Ascomycota, and Blastocladiomycota (Figure 2c). Rozellomycota was the most abundant taxon in A2 (56.6%) and A3 (58.1%), in contrast to that in A1(6.2%) and A4 (16.2%). However,

Chytridiomycota was predominant in A1 (65.4%) and A4 (24.9%) rather than in A2 (7.0%) and A3 (5.3%). Basidiomycota was the third-largest phylum and mainly detected in A4 (29.3%) instead of A1 (5.7%), A2 (8.3%), and A3 (7.2%) (Table S5). Similarly, an evolutionarily clustered heatmap analysis related to the top 30 abundant genera was conducted to illustrate the relative abundance and distribution of microbial communities inhabiting river sediments at the genus level. Fifteen archaeal genera belonging to five phyla (Halobacterota, Crenarchaeota, Euryarchaeota, Thermoplasmatota, Nanoarchaeota) were identified (Figure 3a). The *Methanosaeta*, *Methanoregula*, *Methanosarcina*, *Methanolinea*, and *Candidatus Methanoperedens* in the Halobacterota phylum and *Candidatus Methanomethylicus*, *unidentified Bathyarchaeia* in the Crenarchaeota phylum occupied a major portion of the archaeal ecosystem in A4, while those genera identified in A1 were less abundant. In terms of bacterial and fungal communities, the top 30 abundant genera belonged to 9 and 8 phyla, respectively. The dominant bacterial genera in A1 were mainly assigned to the Proteobacteria phylum, while the dominant genera in A2 and A3 were primarily assigned to the Firmicutes and Proteobacteria phylum. In contrast, the bacterial phyla of the dominant genera in A4 were more diverse (Figure 3b). Several unclassified genera frequently found in the phylum Ascomycota constituted a considerable proportion of the fungal community. Additionally, dominant fungal genera showed a distinctive distribution among different sampling sites (Figure 3c).

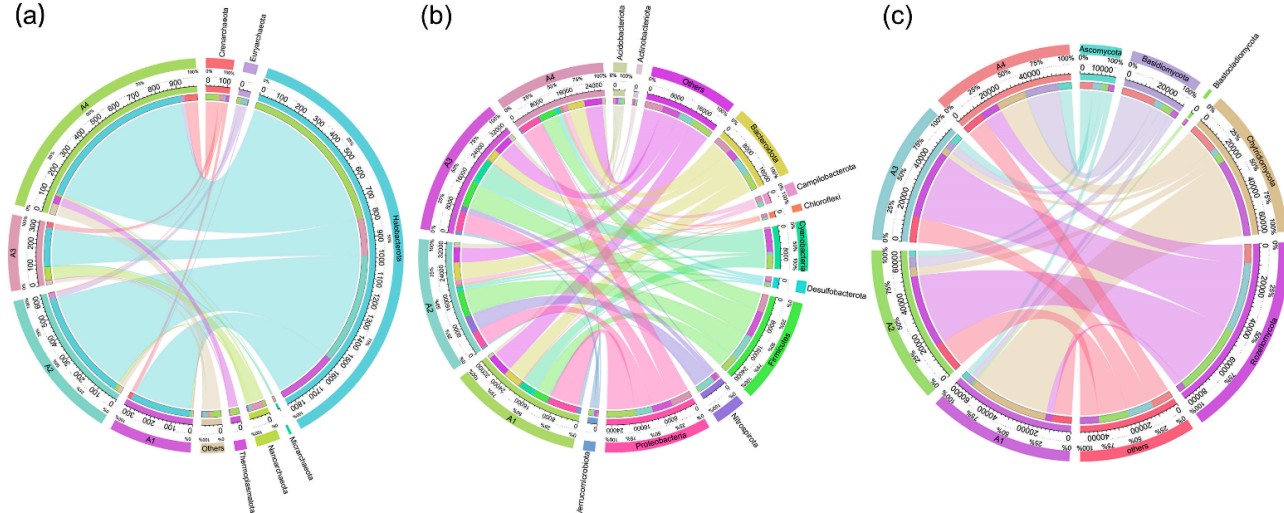

**Figure 2.** Circular representation of archaeal (**a**), bacterial (**b**), and fungal (**c**) communities in sediment samples at the phylum level. Taxa with abundance less than 1% in all samples and sequences that could not be assigned at the phylum level were combined into a group and summarized as "Others". The left part represents the composition of different phyla in each sample, and the length indicates the abundance of species in the sample. The right part represents the proportion of different samples in a specific phylum, and the length shows the distribution of the samples.

### 3.4. Correlation between Microbial Communities and Environmental Variables

Redundancy analysis was performed to interpret the contribution of physicochemical factors on the variation of microbial community structures in sediments. All archaeal genera and the top 30 abundant bacterial and fungal genera were used for RDA, along with the detected physicochemical factors (Figure 4). The first two axes explained 95.17% and 4.52% of archaeal variations (Figure 4a), 63.22% and 30.74% of the bacterial variations (Figure 4b), and 58.67% and 24.54% of the fungal variations (Figure 4c), respectively. The metal contents explained a higher variance in the composition of archaea and bacteria than fungi, followed by salinity and TDS. The content of Al indicated a highly positive correlation with microbial communities in sample A4, especially for the genera *Methanoregula*, *unidentified Basidiomycota* sp., and *unidentified Rhizophydiaceae*. Similarly, the Mn was positively correlated with sample A2, especially for the genera *Methanosaeta*, *unidentified*

*Nitrospiraceae*, and *unidentified Rozellomycota* sp. Depth was the relatively weak determinant factor for explaining the most variations in the community compositions, given by a short projection on the vertical x-axis. For the archaeal community, samples A1 and A2 were negatively correlated with depth. The bacterial and fungal communities in samples A1 and A4 were both positively correlated with depth. In contrast, the bacterial and fungal communities in sample A2 were negatively correlated with depth.

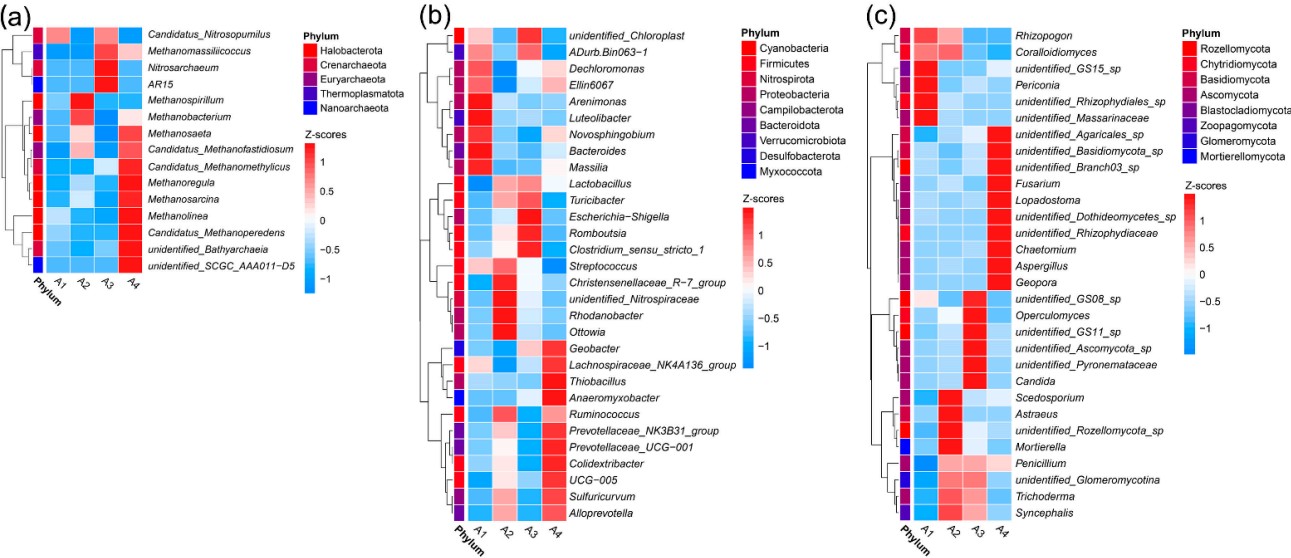

**Figure 3.** Heatmap of the most abundant archaeal (**a**), bacterial (**b**), and fungal (**c**) genera distribution in each sample. The colors in the heatmap represent the relative abundance of the genera normalized by the Z-score. A positive Z-score (red) indicates that the relative percentage of each genus is above the mean, while a negative Z-score (blue) suggests that it is below the mean. Clusters based on taxa along the Y-axis are indicated on the left side of the figure.

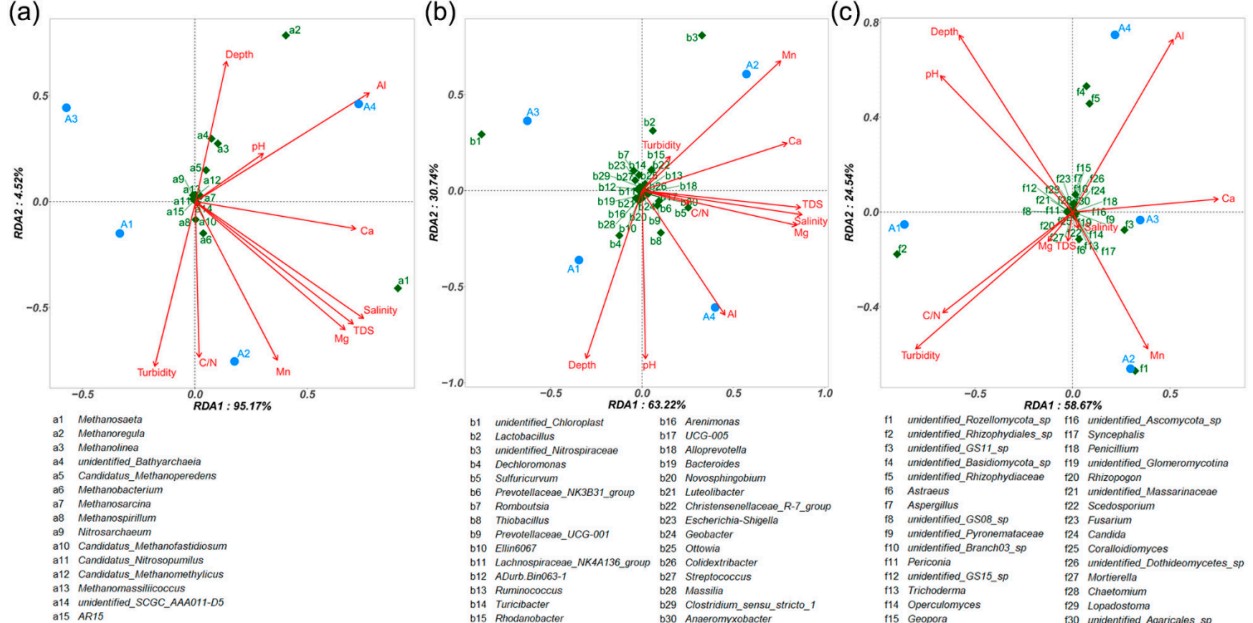

**Figure 4.** Redundancy analysis of sediment samples based on the top abundant archaeal (**a**), bacterial (**b**), and fungal (**c**) genera and physicochemical variables. The values of Axes 1 and 2 are the percentages that can be explained by the corresponding axis. The sediment samples are represented as blue circles. The environmental variables are represented as red arrows, and the genera are represented as green squares.

### 3.5. Functional Predation Analysis

According to the functions predated by the FAPROTAX and FUNGuild database outputs, the top 20 bacterial and fungal functional groups in the core microbiota of river sediments are illustrated in Figure 5. The most dominant bacterial function in all samples was chemoheterotrophy, followed by fermentation. The abundance functional features in A1 that were associated with the biogeochemical cycling of nitrogen (e.g., nitrate reduction, nitrate respiration, and nitrogen respiration) were relatively low in A2, and their proportion gradually increased at the sampling sites of A3 and A4. However, the functional abundance related to carbon metabolism (e.g., methanogenesis, hydrogenotrophic methanogenesis, methylotrophy, and hydrocarbon degradation) was generally low in all the samples. Functions related to sulfide oxidation, such as dark sulfide oxidation and dark oxidation of sulfur compounds, were only found to be abundant in A4 (Figure 5a). Unlike the diversification of bacterial functions, the predicted fungal functions mainly focused on saprotroph, including wood saprotroph, soil saprotroph, and dung saprotroph (Figure 5b). The relative abundance of ectomycorrhizal function was the highest in A2 and the lowest in A3.

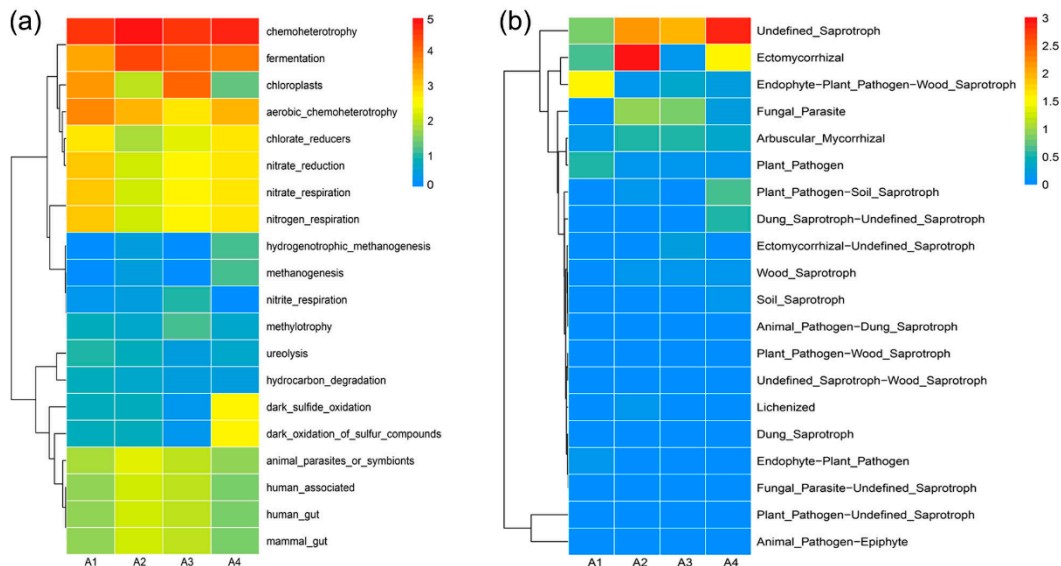

**Figure 5.** Heatmap clustering based on the relative abundance of bacterial (**a**) and fungal (**b**) functional profiles inferred by FAPROTAX and FUNGuild. The heatmap plot depicts the relative percentage of each functional group by color intensity. Y-axes of figures are presented on a log2 scale.

## 4. Discussion

Particle characteristics such as turbidity and total suspended solids have been recommended as indicators when monitoring urban runoff and river pollution [32]. Urban surface runoff with high turbidity impacts the water quality dynamics of downstream ecosystems. In this study, the Beitang River is a polluted urban river [10] with continuously high turbidity (e.g., site A1, 315 ± 11 NTUs) that affects the downstream Xixinghou River and the lives of surrounding residents. As a chemical coagulant, PAC exhibits efficient coagulation performance for removing particles from raw water [14]. It can be seen that the turbidity of the river water began to decrease after PAC treatment (site A2, 280 ± 9.80 NTUs) and was significantly reduced downstream (3.67~5.91 NTUs). Compared with untreated water (site A1), the highest turbidity removal efficiency was 98.1~98.8%, which was consistent with the previous report [33]. The PAC coagulant used in the CF treatment process led to increased metal content in the sediments (site A2), especially Mg, Al, Ca, and Mn, as well as an increase in salinity and TDS. These changes were not caused by differences in sample depth because no strong positive or negative correlations were found between depth and metal content, salinity, or TDS (Figure 4). It was noticed that PAC

coagulant application could accumulate a high concentration of Al downstream of the river. It is known that PAC is less sensitive to temperature and, thus, provides better coagulation at low temperatures [34]. However, a large amount of PAC is still needed to accelerate the formation of flocs and ensure satisfactory coagulation performance because of the special nature of low-temperature water [35]. Excessive use of PAC can easily lead to an increase in residual Al. Although Al is the most widely distributed metal in the environment and the third most abundant element on the Earth, toxicosis occurs when exposed to high levels of Al frequently [36]. Therefore, how to reduce the residual Al content as much as possible when improving the coagulation performance is a practical problem that needs attention. Adjusting the basicity of PAC and using chitosan as coagulant aid are supposed to be useful options for residual Al control when dealing with low-temperature raw water [17]. The increased contents of Ca, Mn, and Mg in the sediments of site A2 are likely caused by the chemical coagulation of PAC. After all, PAC has been proven to be efficient in removing metal contamination [37]. Although high-residue metals in downstream sediments (e.g., site A3) are removed as the river flows into residential areas (e.g., site A4), the TDS, salinity, and metal content will increase again. This might be related to the impact of human activities on nearby communities. As a common anthropogenic alteration, the salinization of rivers and streams caused by human activities has long been acknowledged as a global and growing threat [38].

All calculated alpha diversity estimators (Shannon, Simpson, Chao1, and ACE) indicated that the diversity of the bacterial community was reduced after the application of PAC at sites A2 and A3 (Table S2). The high-residue Al content (Table 1) and the negative correlation between bacterial community and Al content (Figure 4) might be some of the reasons for this phenomenon. Interestingly, the bacterial community in site A4 showed higher alpha diversity and a strong positive correlation with Al content (Table S2 and Figure 4). This was probably due to the high concentration of Ca in site A4 that could ameliorate the toxicity of other metal ions, including Al [39]. In contrast, the fungal community exhibited higher alpha diversity indices at sites A2 and A3 instead of site A1, which probably resulted from the high tolerance of fungi to residual metals [40]. The negative correlation between bacterial diversity and fungal diversity presented in this study is in accordance with a previous report [41]. This has been suggested by the fact that bacteria and fungi play different roles in regulating biochemical conditions in response to environmental changes and maintaining the stability of the ecosystem [42]. At the community level, more research on the relationship between salinity and species richness is needed to identify microbiological characteristics related to salinization resistance or sensitivity and to predict community changes caused by salt pollution.

Although the sampling sites were connected by river flow, the microbial communities in different sampled sediments were different from each other. Distinct variations in sediment microbial communities in spatial shifts can enhance our understanding of the potential ecological effects caused by PAC coagulation. The contributions of dispersal limitation and environmental heterogeneity were proven to be the dominant factor controlling microbial composition variations in river systems [43,44], which was consistent with the apparent shift of microbial communities at the phylum and genus levels in our sampled river sediments.

Compared with bacteria and fungi, the archaeal community takes up a limited proportion of river sedimental microflora. Halobacterota was the most detected archaeal phylum in all river sediments. In general, members in the Halobacterota have a more diverse substrate range, including acetoclastic, hydrogenotrophic, and methylotrophic pathways of methanogenesis and stress-adaptive biodiversity, such as thermophilic, acidophilic, and alkaliphilic tolerance [45], conferring their adaptation to various environmental conditions. The genera *Methanolinea*, *Methanoregula*, *Methanospirillum*, and *Methanosaeta*, belonging to Halobacterota, observed here, showed dramatic variations from upstream (e.g., site A1) to downstream (e.g., site A4) (Figure 3a), suggesting changes in the ecological functions of methanogenesis. Despite the phylogenetic diversity of river sediments spanning six

archaeal phyla in this study, their function was limited to methane cycling, such as the detected *Methanolinea*, *Methanoregula*, *Methanospirillum*, and *Methanosaeta* within Halobacterota, *Candidatus Methanomethylicus* within Crenarchaeota, *Candidatus Methanofastidiosum* and *Methanobacterium* within Euryarchaeota, and *Methanomassiliicoccus* within Thermoplasmatota. Although methanogenesis is an important part of the global biogeochemical engine driving the Earth's energy and carbon cycle, excessive emissions of methane will contribute greatly to global warming and climate change [46]. Therefore, it is important to understand the composition and dynamic changes of methanogenic communities to better limit the harmful effects of methane release.

The bacterial community composition in this study was dominated by the phyla Proteobacteria (15.3~26.8%), Firmicutes (14.5~23.4%), Cyanobacteria (1.71~19.8%), and Bacteroidota (9.57~18.3%). It has been suggested that these major freshwater indigenous phyla may serve as urbanization bioindicators [47] due to their high frequency of detection in the sediments of urban river networks [48–50]. Rozellomycota (6.16~58.1%), Chytridiomycota (5.28~65.4%), and Basidiomycota (5.70~29.3%) were the most abundant fungal phyla in all samples, which was previously found to be abundant in aquatic fungi ecosystems [51,52]. In addition, considerable unclassified bacterial (11.4~16.5%) and fungal (15.6~21.4%) populations were discovered, highlighting the low coverage of currently available 16S rRNA and ITS sequence databases and the highly novel microbial phylogeny that has not been deciphered in aquatic ecosystems. Proteobacteria accounts for the highest proportion of bacterial OTUs in sites A1 (26.8%) and A4 (23.0%), while Firmicutes dominated in sites A2 (23.4%) and A3 (23.2%) (Table S4). Similarly, Rozellomycota was the most dominant fungal phylum in sites A2 (56.6%) and A3 (58.1%), while the most abundant fungal taxa in site A1 was Chytridiomycota (65.4%) and Basidiomycota (29.3%) in site A4 (Table S5). Knowledge about microbial community assembly is essential because ecosystem function is linked with the traits of the taxa comprising that community [53]. For example, the high potential of chloroplast function was observed in sites A3 and A1, where the phylum Cyanobacteria was also relatively abundant, rather than sites A2 and A4 (Figure 5a and Table S4). It seemed likely that most of the top genera were heterotrophic bacteria. Therefore, the highest abundance of predicted function was involved in the metabolic pathways such as chemoheterotrophy, fermentation, and aerobic chemoheterotrophy (Figures 3b and 5a). Unlike the functional diversity of bacteria, all the sediment samples had a large number of fungal OTUs that conferred saprotrophic function (Figure 5b). This observation is consistent with the interpretation that important drivers of plant litter decomposition in streams are a group of freshwater fungi [54].

## 5. Conclusions

This study was conducted to investigate the effect of PAC coagulant on the physicochemical properties and assembly of microbial communities associated with functions in urban river sediments. As a chemical coagulant, PAC exhibited high efficiency in removing particles and reducing the turbidity of urban river water, with a maximum efficiency of 98%. After the addition of PAC coagulant, the TDS, salinity, and residual metals in the sediment increased in an obvious manner. The contents of residual Al in sediments accumulated along the river, ranging from $0.0729 \pm 0.00125$ to $0.599 \pm 0.00275$ mg/L. Compared with the sediments of PAC coagulation in treated and untreated sites, there were obvious changes in the biodiversity of archaea, bacteria, and fungi. The diversity of prokaryotic communities decreased after the application of PAC coagulant, while the diversity of eukaryotic communities increased. Furthermore, the abundance of the core microbiome was also affected. The dominant bacterial phylum obtained by sequencing was affiliated to Proteobacteria (15.3~26.8%), followed by Firmicutes (14.5~23.4%) and Cyanobacteria (1.71~19.8%). These members comprised the functional units that perform specific biogeochemical cycles, such as nitrogen, carbon metabolism, and sulfide oxidation. Rozellomycota (6.16~58.1%), Chytridiomycota (5.28~65.4%), and Basidiomycota (5.70~29.3%) accounted for the highest proportion of fungal taxa. The predicted fungal functions mainly focused

on saprotroph, including wood saprotroph, soil saprotroph, and dung saprotroph. The archaeal community occupied a limited proportion of river sedimental microflora, mainly associated with methanogenesis. The variation of core community was closely related to the environmental factors of aquatic ecosystems, such as salinity, TDS, and metal content, suggesting that the distribution of particular taxa was driven by their metabolic capabilities in response to environmental factors. Overall, we determined that PAC coagulant had a profound impact on the microbial diversity and abundance of urban river sediments. These differences were predominantly due to changes in the relative abundances of a shared core community. While there is still a lot of work to be done in assessing the driving factors of microbial community structure and function, this work promotes the understanding of human impacts on freshwater micro-ecosystems.

**Supplementary Materials:** The following are available online at https://www.mdpi.com/article/10.3390/w13131791/s1, Figure S1. Rarefaction curves of the sediment samples from bacteria (a) and fungi (b) at 97% similarity grouping of the 16S V3-V4 and ITS1 sequences. The vertical axis shows the number of OTUs expected to be found after sampling the numbers of tags shown on the horizontal axis. Lines of different colors represent different sediment samples. Table S1. Summary statistics of the sequencing results for the bacterial and fungal communities. Table S2. Calculated alpha diversity indices of the bacterial and fungal communities. Table S3. Relative abundance of archaeal phylum in different samples. Table S4. Relative abundance of bacterial phylum in different samples. Table S5. Relative abundance of fungal phylum in different samples.

**Author Contributions:** Conceptualization, S.L. and X.Z.; validation, C.W.; formal analysis, S.L.; investigation, X.Z.; resources, X.Z.; writing—original draft preparation, S.L.; writing—review and editing, C.W.; visualization, S.L.; funding acquisition, C.W. All authors have read and agreed to the published version of the manuscript.

**Funding:** This research was funded by the China Postdoctoral Science Foundation (2020M672048) and the Qingdao Postdoctoral Application Research Project.

**Institutional Review Board Statement:** Not applicable.

**Informed Consent Statement:** Not applicable.

**Data Availability Statement:** The data presented in this study are available on request from the corresponding author.

**Acknowledgments:** The authors thank Chengjia Zhang and Nannan Dong from Core Facilities for Life and Environmental Sciences (State Key Laboratory of Microbial Technology, Shandong University) for assistance in elemental analysis.

**Conflicts of Interest:** The authors declare no conflict of interest.

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
