# Peer review of "Application of Polyaluminium Chloride Coagulant in Urban River Water Treatment Influenced the Microbial Community in River Sediment"

_water, doi:10.3390/w13131791_

Round 1
Reviewer 1 Report
Dear authors, please find the detailed comments in the attached file.

Reviewer 2 Report
This is a very timely and informative paper. However, why was only one sample site (A1) selected in the Beitang River?
The results section under physiochemical properties needs to be more detained, use specific numbers or something where the reader does not have to always look at the tables to follow. Lack of details in the paragraph interrupts the flow. For instance the author says the River was turbid. What exactly does that mean especially when the turbidity was actually measured.
Line 97…”using” instead of “used”
Line 147…Delete “Besides” and the “,” after reduced.
Line 151…Metals increased to what. The author could have given a range of values within the paragraph. Again, this makes the reader generalize the values and then have to go find in the table.
Could the supplementary table for the alpha diversity be in the paper or is there space limitation?
Line 241….Begin the sentence with full name for RDA
Overall the paper is detailed with sufficient analysis. There is just the question about the sampling the design of not having more than one sampling site from the Beitang River. Also, just making the results sections more detailed so that the paper flows better.
Reviewer 3 Report
The methodology requires minor corrections. Mark the places of coagulant dosing on the map. Why is point A1 taken as a reference?
Whether the differences in the depth of the river cannot affect the parameters tested, both physico-chemical and microbiological - please add the results in the discussion.
